# Mutations and Copy Number Alterations in *IDH* Wild-Type Glioblastomas Are Shaped by Different Oncogenic Mechanisms

**DOI:** 10.3390/biomedicines8120574

**Published:** 2020-12-07

**Authors:** Ege Ülgen, Sıla Karacan, Umut Gerlevik, Özge Can, Kaya Bilguvar, Yavuz Oktay, Cemaliye B. Akyerli, Şirin K. Yüksel, Ayça E. Danyeli, Tarık Tihan, O. Uğur Sezerman, M. Cengiz Yakıcıer, M. Necmettin Pamir, Koray Özduman

**Affiliations:** 1Department of Biostatistics and Medical Informatics, School of Medicine, Acibadem Mehmet Ali Aydinlar University, Istanbul 34752, Turkey; ege.ulgen@live.acibadem.edu.tr (E.Ü.); sila.karacan@live.acibadem.edu.tr (S.K.); Umut.gerlevik@live.acibadem.edu.tr (U.G.); Ugur.Sezerman@acibadem.edu.tr (O.U.S.); 2Department of Medical Engineering, Faculty of Engineering, Acibadem Mehmet Ali Aydinlar University, Istanbul 34752, Turkey; OZGE.CAN@acibadem.edu.tr; 3Department of Genetics, School of Medicine, Yale University, New Haven, CT 06520, USA; kaya.bilguvar@yale.edu; 4Yale Center for Genome Analysis, West Haven, CT 06516, USA; 5Izmir Biomedicine and Genome Center (IBG), Izmir 35340, Turkey; yavuz.oktay@ibg.edu.tr; 6Department of Medical Biology, Faculty of Medicine, Dokuz Eylül University, Izmir 35340, Turkey; 7Department of Medical Biology, School of Medicine, Acibadem Mehmet Ali Aydinlar University, Istanbul 34752, Turkey; Cemaliye.Boylu@acibadem.edu.tr (C.B.A.); sirin.yukselkilicturgay@acibadem.edu.tr (Ş.K.Y.); 8Department of Pathology, School of Medicine, Acibadem Mehmet Ali Aydinlar University, Istanbul 34752, Turkey; Ayca.Ersen@acibadem.edu.tr; 9Neuropathology Division, Department of Pathology, School of Medicine, University of California San Fransisco (UCSF), San Francisco, CA 94143, USA; Tarik.Tihan@ucsf.edu; 10Department of Molecular Biology, School of Arts and Sciences, Acibadem Mehmet Ali Aydinlar University, Istanbul 34752, Turkey; mcengiz.yakicier@acibadem.com.tr; 11Department of Neurosurgery, School of Medicine, Acibadem Mehmet Ali Aydinlar University, Istanbul 34752, Turkey; Necmettin.Pamir@acibadem.edu.tr

**Keywords:** glioma, mutational signatures, DNA repair, exome sequencing

## Abstract

Little is known about the mutational processes that shape the genetic landscape of gliomas. Numerous mutational processes leave marks on the genome in the form of mutations, copy number alterations, rearrangements or their combinations. To explore gliomagenesis, we hypothesized that gliomas with different underlying oncogenic mechanisms would have differences in the burden of various forms of these genomic alterations. This was an analysis on adult diffuse gliomas, but *IDH*-mutant gliomas as well as diffuse midline gliomas *H3*-K27M were excluded to search for the possible presence of new entities among the very heterogenous group of *IDH*-WT glioblastomas. The cohort was divided into two molecular subsets: (1) Molecularly-defined GBM (mGBM) as those that carried molecular features of glioblastomas (including *TERT* promoter mutations, 7/10 pattern, or *EGFR*-amplification), and (2) those who did not (others). Whole exome sequencing was performed for 37 primary tumors and matched blood samples as well as 8 recurrences. Single nucleotide variations (SNV), short insertion or deletions (indels) and copy number alterations (CNA) were quantified using 5 quantitative metrics (SNV burden, indel burden, copy number alteration frequency-wGII, chromosomal arm event ratio-CAER, copy number amplitude) as well as 4 parameters that explored underlying oncogenic mechanisms (chromothripsis, double minutes, microsatellite instability and mutational signatures). Findings were validated in the TCGA pan-glioma cohort. mGBM and “Others” differed significantly in their SNV (only in the TCGA cohort) and CNA metrics but not indel burden. SNV burden increased with increasing age at diagnosis and at recurrences and was driven by mismatch repair deficiency. On the contrary, indel and CNA metrics remained stable over increasing age at diagnosis and with recurrences. Copy number alteration frequency (wGII) correlated significantly with chromothripsis while CAER and CN amplitude correlated significantly with the presence of double minutes, suggesting separate underlying mechanisms for different forms of CNA.

## 1. Introduction

Gliomas vary considerably in their phenotype, biology, clinical behavior and response to treatment [1]. Various distinct tumor entities, including astrocytoma, oligodendroglioma and glioblastoma (GBM), were initially defined by morphological criteria and further characterized by large-scale molecular and genetic studies [2,3,4,5]. Each tumor type has a specific molecular landscape with distinctive methylation profiles indicating their cell of origin and genomic alterations defining oncogenic programs [6,7]. Such divergent molecular-genetic landscapes imply that the causative mechanisms may be different, but little is known on the subject [8]. Large-scale genomics analyses exploring mutational signatures indicated that clock-like mutational processes (spontaneous deamination of methylcytosine) and temozolomide-related signatures were the predominant mechanisms in gliomas [9]. Other studies have also provided evidence that DNA repair deficiency was a central theme in gliomagenesis [10,11].

Variations in the nucleotide sequence are not the only form of genetic alterations. Other forms like alteration in chromosome number (aneuoploidy) and structure (copy-number alterations, inversions and re-arrangements) also play major roles in shaping the cancer genome. These alterations are caused by a different spectrum of mechanisms acting at different stages of gliomagenesis in distinct glioma entities [12,13,14,15].

Isocitrate dehydrogenase (*IDH*) enzymes participate in a variety of metabolic mechanisms, such as Krebs cycle, glutamine metabolism, lipogenesis, redox regulation, and cellular homeostasis, by catalyzing the oxidative decarboxylation of isocitrate. Previous studies revealed that mutations of *IDH* genes are frequently observed in several human malignancies, including gliomas, and that they play a potential role in oncogenesis [16]. *IDH* mutations are recognized in the majority of the lower-grade gliomas and they are associated with more favorable outcome than *IDH* wild-type. *H3*-K27M mutations were first recognized in pediatric diffuse intrinsic pontine gliomas, but thereafter, they have been observed in midline gliomas in adults [17]. Studies on *H3*-K27M-mutant tumors indicate that *H3*-K27M mutant gliomas were diagnosed at an earlier age and have poor prognosis [18,19,20].

In this study, we chose to analyze the heterogenous group of *IDH*-WT diffuse gliomas, which are probably made up of many different entities. Other well-characterized diffuse gliomas such as “*IDH*-mutant gliomas” (astrocytomas and oligodendrogliomas) and “diffuse midline gliomas *H3*-K27M mutant” were deliberately excluded [1]. We studied the oncogenic processes of the cohort indirectly by quantifying the corresponding genetic alterations that they cause and subsequently correlating these findings with direct measurements of several oncogenic processes. The aim of this study was to analyze the burden of genetic alterations in different *IDH*-WT entities. Our hypothesis was that the burden of various genetic alterations in these different *IDH*-WT glioma entities would reflect variations in driving genomic alterations which acted upon them.

## 2. Materials and Methods

### 2.1. Patients and Tumor Samples

Thirty-nine adult patients (24 male and 15 female, median age = 51 (range = 28–76)) who were operated on or underwent stereotactic biopsy for diffuse gliomas were included. *IDH*-mutant gliomas (astrocytomas and oligodendrogliomas) as well as diffuse midline gliomas *H3*-K27M mutant were excluded. 45 tumor samples from 39 patients were studied, including 37 primary tumors and 8 recurrences after radiochemotherapy (radiotherapy and temozolomide). Characteristics of patients and tumors are presented in Table 1. All patients were informed about whole exome sequencing (WES) testing and provided written consent. The study was approved by Acıbadem Mehmet Ali Aydınlar University institutional review board (ATADEK-2018/7, 17.05.2018).

### 2.2. Pathology and Molecular Subsets

All pathological specimens were retrospectively reviewed by a single neuropathologist (A.E.D.). Molecular markers (including *TERT* promoter mutations) were determined using WES and/or Sanger sequencing and/or fluorescent in situ hybridization. Molecular subsets were determined as follows: *IDH*-wild-type gliomas with “*TERT* promoter mutations” and/or “*EGFR* amplifications” and/or “chromosome 7 amplifications and chromosome 10 loss” were classified as “molecularly-defined glioblastoma (mGBM)” [21,22]. Other less common and less well-defined *IDH*-WT gliomas were grouped as “other diffuse gliomas” (“Others”), including 4 hemispheric high-grade gliomas which were *SETD2*-mutant, 1 diffuse glioma which was *H3*-G34-mutant and 1 anaplastic astrocytoma with piloid features (as confirmed by methylation profiling, Data not provided) (Table 1) [23].

*IDH*-WT gliomas are a very heterogeneous group of tumors, likely containing entities which remain to be identified. Therefore, we classified *IDH*-WT gliomas which carried the generally accepted molecular markers of glioblastoma as mGBM [21] and the remaining as “Others”. This grouping of *IDH*-WT gliomas was performed as these are different entities with different molecular features as well as different clinical characteristics. Because we only included *IDH*-WT gliomas in this study, we do not report any comparison between primary versus secondary GBM.

### 2.3. Whole Exome Sequencing, Pre-Processing and Variant Calling

DNA was extracted from snap-frozen tumor and peripheral venous blood samples using the DNeasy Blood and Tissue Kit (QIAGEN, Hilden, Germany). Sequencing of the libraries were performed on Illumina (San Diego, California, USA) HiSeq instruments using paired-end reads. FASTQ data are available under the European Genome-Phenome Archive (https://ega-archive.org) accession EGAD00001004144. We achieved mean target coverage of 207.27 and 126.25, for tumors and matching blood samples, respectively. Detailed sequencing quality information, including exome capture kit information, is provided in Appendix A.

The reads were aligned to the reference genome (UCSC hg19 assembly) using BWA-MEM (version 0.7.17-r1188) [24]. The mapped reads were cleaned with Picard-CleanSam (Picard version 2.21.6-SNAPSHOT http://broadinstitute.github.io/picard/; Cambridge, Massachusetts, USA). Cleaned reads were sorted and mate information was fixed using Picard-FixMateInformation. PCR-Duplicates were marked using Picard-MarkDuplicates. Base quality scores were recalibrated using the Genome Analysis Toolkit (GATK, version 4.1.4.0; Cambridge, Massachusetts, USA). Somatic Single Nucleotide Variation (SNV) and insertion/deletion (indel) calling was performed using GATK-MuTect2. Somatic copy number alterations (SCNAs) were identified using ExomeCNV [25]. Somatic structural variations were detected using DELLY [26].

### 2.4. Metrics

A summary of the 5 quantitative metrics and 4 parameters that explored underlying oncogenic mechanisms are presented in Table 2. Analyses were performed using R (https://www.R-project.org/, Vienna, Austria).

1. SNV burden was defined as the number of somatic SNVs in the coding region per megabase. After (a) keeping variants with variant allele frequency (VAF) > 5% and (b) keeping variants with a sequence depth > 20X in the tumor and > 10X in the normal sample, somatic SNV burden was calculated as:(1)$$\frac{\;\#\;SNVs}{exome\;length\left(Mb\right)}$$

2. Indel burden was defined as the number of somatic indels in the coding region per megabase. After filtering using the same criteria for SNVs, somatic indel burden was calculated as:(2)$$\frac{\;\#\;indels}{exome\;length\left(Mb\right)}$$

3. The weighted Genome Instability Index (wGII) was used to determine the fraction of the exome exhibiting copy number alterations [15]. The fractions of altered (defined as |log_2_ ratio| > 0.25) segments over the total size of the regions captured by the exome kit were calculated for each autosomal chromosome and aggregated via the overall average to eliminate the bias induced by variation in chromosomal sizes.

4. Chromosomal Arm Event Ratio (CAER) was used to determine chromosomal-arm-level SCNAs. For each chromosomal arm, the weighted arithmetic mean of the Tumor/Normal ratios of all segments within the arm was calculated and log_2_-transformed. If an arm had a |weighted-mean-log_2_-ratio| > 0.25, a chromosomal-arm-level SCNA was determined (Appendix A). CAER was determined as the ratio:(3)$$\frac{\;\#\;arms\;with\;SCNA}{\#\;all\;autosomal\;arms}$$

5. Copy-number amplitude was defined as the highest copy-number observed [27].

6. Chromothripsis events were determined using CTLPScanner which detects the copy-number change clusters via sliding windows, calculating a likelihood ratio for each window [28]. Only autosomal chromosomes were used for assessment.

7. Double minutes (DMs) were detected as previously described [29]. Firstly, high-level copy-number segments with Tumor/Normal ratio ≥ 5 were determined. Possible double minutes were determined if (a) the sample contained multiple distinct high-level copy-number segments, at least one of which was overlapping an oncogene or (b) there was one distinct high-level copy-number segment containing ≥ 1 oncogene, with length > 1 Mb and with an associated structural variation.

8. Microsatellite Instability (MSI) status of each tumor was predicted using the tool MSIpred, which uses 22 somatic mutational features to predict MSI via a support vector machine model [30].

9. Brain tumor-specific mutational signatures within each tumor were determined using a web-based tool (https://signal.mutationalsignatures.com/) [11]. For high confidence, only signatures with contribution ≥ 10% were accepted.

There was no significant difference in any metrics between Formalin-Fixed Paraffin-Embedded (FFPE) and fresh-frozen tumor samples (LiN2) (Appendix A).

### 2.5. The Cancer Genome Atlas Pan-Glioma Data

The current cohort consisted of cases where WES was performed with clinical intent, introducing a selection bias. Therefore, the findings were validated using The Cancer Genome Atlas (TCGA) pan-glioma study [7]. Only cases with known *TERT* promoter mutation status were used to yield comparable findings.

### 2.6. Association of Somatically Mutated Genes with Metrics and Pathway Enrichment Analyses

An SNV/indel was defined as “high-impact” if its VAF > 5% and its classification was one of: “Frame_Shift_Del”, “Frame_Shift_Ins”, “Splice_Site”, “Translation_Start_Site”, “Nonsense_Mutation”, “Nonstop_Mutation”, “In_Frame_Del”, “In_Frame_Ins”, “Missense_Mutation”. For each gene with a high-impact somatic SNV/indel, Wilcoxon rank-sum test was performed to detect any difference of metrics between mutated and non-mutated tumors. Hence, genes associated with each metric were obtained. Next, pathway enrichment analyses of the associated genes were conducted using pathfindR [31].

## 3. Results

For analyses, 45 *IDH*-WT diffuse glioma tumor specimens from 39 patients were used, classified as mGBM (n = 37, 82.22%) or “Others” (n = 8, 17.78%) (Table 1). There were 37 (82.22%) primary and 8 (17.78%) recurrent tumors. When only primary cases were considered (n = 37), 31 were mGBMs and 6 were “Others”.

For validation, we analyzed the TCGA pan-glioma cohort [7]. The *IDH*-WT diffuse gliomas in the TCGA pan-glioma cohort (with known *TERT* promoter mutation status) consisted of 83 cases, 71 mGBMs (85.54%) and 12 others (14.46%), all primary tumors.

For primary gliomas, the median SNV burden was 3.38 (range = 0.48–55.53), the median indel burden was 0.38 (range = 0.09–2.55), the median wGII, measuring SCNA frequency, was 0.28 (range = 0.02–0.88), the median CAER, measuring aneuploidy degree, was 0.16 (range = 0–0.68) and the median copy-number (CN) amplitude was 15 (range = 4–149). Fifteen primary cases (40.54%) had CT events, and sixteen primary samples contained putative DMs (43.24%). A total of 44 oncogenes were detected in putative DMs across the 16 samples with at least one oncogene identified in every DM. *EGFR* and *SEC61G* (both observed in n = 8, 50%) were the most frequent, followed by *VOPP1* (n = 4, 25%), *MDM2*, *CDK4* and *AGAP2* (each n = 3, 18.75%) (Appendix A). No kataegis event was observed in the current cohort nor the TCGA cohort. The MSI prevalence was low (n = 4, 8.89% in the current and n = 2, 2.41% in the TCGA cohort).

### 3.1. Comparison of Metrics between Molecular Subsets

We compared the metrics between the molecular subsets in primary tumors. In the current cohort, the median SNV burden values of mGBM (3.38/Mb) and “Others” (3.59/Mb) were similar (*p* = 0.89, Figure 1A). In the TCGA cohort, mGBM (1.34/Mb) had higher SNV burden compared to “Others” (0.1/Mb, *p* < 0.001). There was no significant difference in indel burden of different molecular subsets in neither the current nor the TCGA cohort (*p* = 0.77 and *p* = 0.48 respectively, Figure 1B). In the current cohort, mGBM had higher median wGII (0.29) compared to “Others” (0.13, *p* = 0.035, Figure 1C). The same was observed in the TCGA cohort: mGBM had higher median wGII (0.19) than “Others” (0.01, *p* < 0.001). In the current cohort, there was no significant difference in median CAER between the subsets (*p* = 0.086, Figure 1D). In the TCGA cohort, mGBM had higher median CAER (0.41) compared to “Others” (0.33, *p* = 0.011). In the current cohort, mGBM had higher median CN amplitude (29) compared to “Others” (4, *p* = 0.0016, Figure 1E). In the TCGA cohort, again, mGBM had higher median CN amplitude (17) compared to “Others” (5, *p* < 0.001).

Although it may be expected to observe higher SCNA-associated metric levels in mGBMs (due to chr7 gains and chr10 losses), it is important to keep in mind that these metrics are global, assessing all autosomal SCNA events. Therefore, they are expected to be affected little by the canonical chr7 gains and chr10 losses.

We next investigated the correlations between the metrics (Figure 2). Hierarchical clustering based on the correlations yielded 2 clusters: (1) SNV burden and indel burden, associated with mutational processes, i.e., metrics/processes associated with changes in nucleotide sequence, and (2) CN amplitude, DM, CT, wGII and CAER, associated with SCNA-related mechanisms, i.e., metrics/processes associated with changes in chromosomal copy-number/structure.

### 3.2. Pathway Enrichment Analysis of Metric-Associated Somatic Variants

To investigate the possible mechanisms underlying each metric, we firstly examined the genes associated with each metric. Through Wilcoxon rank-sum tests, 1050, 886, 92, 36 and 19 genes with somatic SNV/indels were found to be significantly associated with SNV burden, indel burden, wGII, CAER and CN amplitude, respectively (Appendix A). Next, pathway enrichment analyses were performed using the associated genes for each metric. As a result, 72, 60, 7, 1 and 4 pathways were found to be enriched for SNV burden, indel burden, wGII, CAER and CN amplitude, respectively. Pathways distinct to each metric and those at each intersection of the enrichment results are presented in Figure 3. There were 19 enriched pathways specific to SNV burden, 10 were specific to indel burden and 4 were specific to wGII. SNV burden and indel burden shared 45 common enriched pathways. SNV burden and wGII had 2 common enriched pathways. SNV burden and CAER shared 1 pathway. SNV burden, indel burden and CN amplitude shared 4 common enriched pathways. SNV burden, indel burden and wGII had 1 common enriched pathway. Of note, the “Mismatch repair” pathway was significantly enriched only for the SNV burden metric.

### 3.3. Correlation of Metrics with Chromothripsis and Double Minutes

In primary tumors, CT events were most frequently observed within chr17 (n = 11, 19.3% of total events), followed by chr1 (n = 8, 14.04%) and chr16 (n = 6, 10.53%), while DM events were most frequent in chr7 (n = 9 DMs, 50% of total), chr12 (n = 3, 16.67%) and chr1 (n = 2, 11.11%) (Figure 4A). CT was associated with higher wGII (*p* = 0.0014) (Figure 4B). The proportion of cases with at least one putative double minute chromosome was not different between cases harboring CT and not harboring CT (*p* = 1). Harboring a DM was associated with higher CAER (*p* = 0.021) and CN amplitude (*p* < 0.001) (Figure 4C).

### 3.4. Associations of Metrics with Age and with Recurrences

Next, correlation of each metric with age at diagnosis was investigated. To remove any confounding effects of molecular class and recurrence, only primary mGBMs were analyzed for both the current and TCGA cohorts (Figure 5). This analysis yielded that SNV burden was positively correlated with age at diagnosis (R = 0.32, *p* = 0.079 for the current cohort, R = 0.25, *p* = 0.037 for the TCGA cohort, Figure 5A). Indel burden, wGII, CAER and CN amplitude displayed no significant correlation with age at diagnosis (Figure 5B–E). In both the current and TCGA cohorts, there was no significant difference in age at diagnosis between cases with CT and without (*p* = 0.48 and *p* = 0.5 for the current and TCGA cohorts, respectively). There was no difference in age at diagnosis between cases with DMs and without (*p* = 0.75).

To evaluate how metrics differ in primary vs. recurrent tumors, first, we compared primary tumors (n = 37) and recurrent tumors (n = 8). SNV and indel burden were significantly higher in recurrent tumors (*p* = 0.0066 and *p* = 0.0057, respectively), whereas wGII, CAER and CN amplitude displayed no significant difference (Figure 6A). There was no significant difference between the proportions of primary and recurrent tumors with and without CT (χ^2^
*p* = 0.67, Figure 6B), 43.24% of primary tumors had DMs compared to 25% of recurrent tumors (χ ^2^
*p* = 0.58, Figure 6C) and 2 primary tumors (5.71%) and 2 recurrent tumors (25%) were predicted to have MSI (χ ^2^
*p* = 0.28, Figure 5D).

### 3.5. Correlation of the Metrics with Mutational Signatures

The most frequent detected central nervous system (CNS)-associated mutational signature in primary tumors was the clock-like signature CNS_B, associated with the deamination of 5-methylcytosine to thymine (n = 33, 88.19%, Figure 7A). There was no association of any signatures with molecular subsets or metrics (Appendix A). When signature contributions were compared between all primary and recurrent tumors, CNS_B, the clock-like signature, was found to be lower in recurrent tumors, whereas CNS_E, associated with mismatch repair, was found to be significantly higher in recurrent tumors (*p* < 0.001, Figure 7B).

## 4. Discussion

### 4.1. Rationale for the Study

Gliomas are a heterogeneous tumor group, consisting of various entities such as “*IDH*-mutant astrocytomas”, “*IDH*-mutant, 1p/19q-co-deleted oligodendrogliomas”, “*IDH*-wild-type GBM” and “diffuse midline gliomas H3 K27M mutant”, which are being characterized in ever increasing detail [1]. These entities differ in demographics, histopathology, molecular markers, clinical behavior, treatment response and outcome [1,7,32,33]. Genetic and epigenetic landscapes are also divergent [6,7]. Even determinants of genetic inheritance are dissimilar [34,35]. Therefore, it would not be irrational to think that each tumor type is formed by different oncogenic processes. Oncogenic processes leading to the observed genetic alterations in gliomas are not well-characterized, but the consequent alterations can be readily quantified. Various forms of genetic alterations exist [14,15]: some affect the genetic sequence (e.g., mutations), others affect the karyotype, with some events resulting in abnormal number of chromosomes (aneuploidy) and others changing the structure of chromosomes (e.g., copy-number alterations, re-arrangements or loss of heterozygosity). These genetic alterations differ in underlying mechanisms [14]. To evaluate the effect of mechanisms underlying molecular subsets of adult diffuse gliomas, we quantified the burden of various genetic alterations. Correlations among metrics indicated that SNV and indel burdens clustered together, whereas metrics of chromosome number/structure formed another cluster (Figure 2). SNV burden increased significantly with advancing age in both the current and TCGA cohorts (Figure 5). No significant correlation with age was noted for the chromosome number/structure-related metrics. Together, these findings may indicate that mutations and alterations of chromosome number/structure are caused by different mechanisms and have different dynamics in gliomas.

### 4.2. Molecular Subsets of IDH-WT Glioblastomas Differ in Genomic Alteration Burden

Molecularly-defined glioblastomas (mGBM) and “Others”, which consisted of diffuse gliomas with no commonly accepted canonical markers, were significantly different in all SNV (only in the TCGA validation cohort) and CNA metrics but the indel burden was comparable (Figure 1). Some discrepancies between our cohort and the TCGA for some metrics may have resulted from selection bias or the size of the cohort.

### 4.3. Chromothripsis and Double Minute Events May Be Drivers of Copy-Number Alterations in IDH-WT Glioblastoma

Little is known about the mechanisms that alter chromosome number/structure in gliomas. Some copy number gains (chr7, chr19, chr20) are early, clonal events in GBM [12]. In addition to the canonical chr7 gains and chr10 losses, other chromosomal arm events were observed scattered throughout the genome in GBMs (Appendix A).

In this study, chromothripsis was found to be a mechanism associated with altered chromosome number and structure. It is characterized by rapid and massive but localized accumulation of chromosomal re-arrangements resulting from nondisjunction events during mitosis [36]. Previously, chromothripsis was reported in 84% of GBM cases [13]. In the current study, chromothripsis events were observed in over one-third of primary tumors (Figure 4). Copy-number alteration frequency (wGII) was significantly higher in cases exhibiting chromothripsis, but SNV or indel burden values were comparable. This may indicate that chromothripsis is a driver of structural variations but not mutations in gliomas. The chromothripsis events were most commonly observed in chr17, chr1 and chr16 (but not in chr7 or in chr10) (Figure 4). These findings together hint that there are multiple mechanisms leading to aneuploidy in GBM and that chromothripsis is a late event.

Double minutes (DM) are small fragments of extrachromosomal DNA, formed via the circularization of highly amplified, double-stranded DNA mostly containing oncogenes [37]. DM was observed in over one-third of cases and the oncogenes contained herein (*EGFR*, *MDM2*, *CDK4*) were consistent with previous reports [29,38]. Tumors with DM displayed higher CAER and CN amplitude, hinting to a role in driving structural variations (Figure 4).

Other possible mechanisms driving structural variations may include chromosomal instability (CIN), break-fusion-bridge (BFB) cycles or kataegis. In this cohort, the frequency of copy-number alterations (wGII), the degree of aneuploidy (CAER) and copy-number amplitude remained fairly constant over increasing age at diagnosis and at recurrences (Figure 5 and Figure 6), which is not consistent with CIN, a continuous process that would result in ever-increasing extent and complexity of copy-number events. Kataegis events, which are associated with *APOBEC* mutagenesis, were not observed in our cohort nor the TCGA. Another mechanism of interest in *TERT* promoter mutant gliomas is occurrence of BFB cycles, which create chromosomal instability until the acquisition of telomerase activity, however the current study was limited due to lack of a reliable bioinformatics tool for its detection from WES data [32].

### 4.4. Mismatch Repair Deficiency Is Likely a Major Driver of Mutational Burden in Gliomas

A pathway enrichment analysis based on the 5 quantitative genomic alteration metrics indicated that genes associated with SNV burden were enriched for mismatch repair (MMR) deficiency. MMR is a highly conserved biological pathway that plays a key role in maintaining genomic stability and MMR-deficiency is a known topic in gliomas. We previously showed that diffuse gliomas had a high incidence of both familial-inherited and somatically gained MMR-deficiency [10]. Several studies also indicated that MMR-deficiency results in an increase in mutational burden [10,39,40,41]. We also observed substantial increases in SNV burden and indel burden at recurrence after radiochemotherapy (Figure 6). In parallel with previous studies, this substantial increase in mutational burden was associated with significantly higher weight of MMR-deficiency-associated mutational signature CNS_E (Figure 7) and with a trend towards higher incidence of MSI (Figure 6). Newly acquired MMR gene mutations were shown to lead to temozolomide resistance [42,43,44]. Temozolomide-induced damage in cells with MMR-deficiency were shown to be the mechanism leading to a post-treatment hypermutated phenotype [45]. In contrast, similar levels of copy-number-related metrics and no signs of chromosomal instability were noted at recurrence after radiochemotherapy (Figure 6). The prevalence of chromothripsis or DM were also not significantly different in post-treatment recurrences. These indicate that MMR is a major driver of new mutations over time and with recurrences after radiochemotherapy.

### 4.5. Limitations and Future Prospects

The cancer genome has a complex nature and the current findings represent most likely only a detail in the gliomagenesis. Furthermore, being a pure bioinformatic analysis, the current work points to associations but does not provide mechanistic analysis of underlying mechanisms. Also, as it is a whole-exome analysis-based work; therefore, various other forms of genetic alterations including rearrangements, intratumoral heterogeneity or epigenetic changes were not addressed. The current analysis is also limited by small sample sizes (resulting in some singular entities), decreasing statistical power. More comprehensive analyses on larger cohorts can advance our understanding of gliomagenesis and point to tumor vulnerabilities. The associations identified in this study should be further investigated to identify and experimentally prove any underlying cause–effect relationship.

## 5. Conclusions

Taken together, these findings support the notion that single nucleotide variations and copy number alterations are driven by separate mechanisms and that the cancer genome in different molecular subsets of *IDH*-WT glioblastomas diverge in the composition of distinct genetic alterations. We hope that this work contributes to the deeper understanding of the tumor biology underlying *IDH*-WT tumor entities and will allow for better characterization of these tumors to better understand clinical behavior and eventually for developing novel treatment strategies.

## Figures and Tables

**Figure 1 biomedicines-08-00574-f001:**
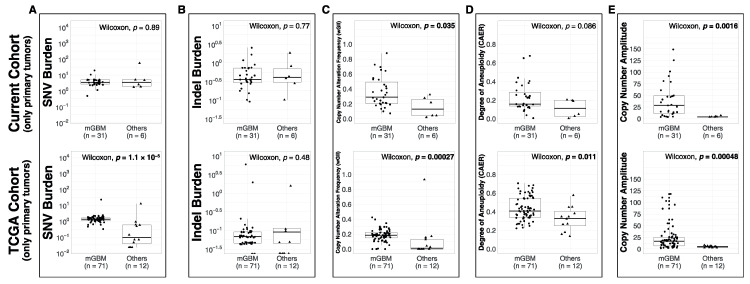
Molecular subsets of *IDH*-WT gliomas differ significantly in their SNV burden, wGII (copy number alteration frequency), CAER (degree of aneuploidy) and copy-number amplitude but not for indel burden. (**A–E**) Distributions of the 5 quantitative metrics in the 2 molecular subsets. The upper row displays findings in only primary cases of this cohort and the lower row displays findings in the TCGA pan-glioma cohort. Bold *p* values indicate statistical significance (*p* < 0.05).

**Figure 2 biomedicines-08-00574-f002:**
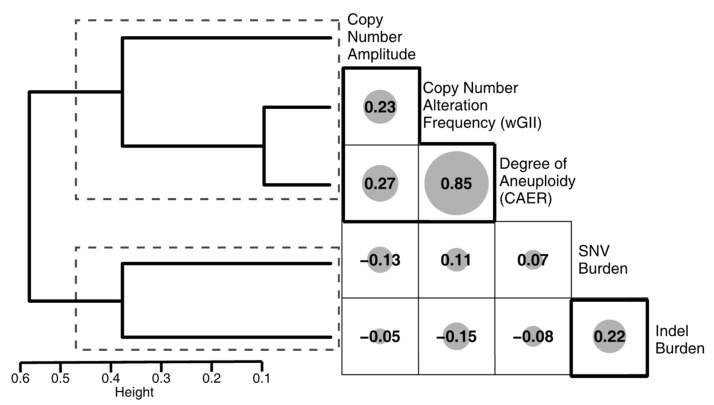
Metrics associated with mutational processes and copy-number-associated processes for 2 distinct clusters. Correlogram of both quantitative and qualitative metric, sizes indicate the |correlation coefficient|. The hierarchical clustering dendrogram is displayed on the left, the identified clusters are indicated by dashed rectangles.

**Figure 3 biomedicines-08-00574-f003:**
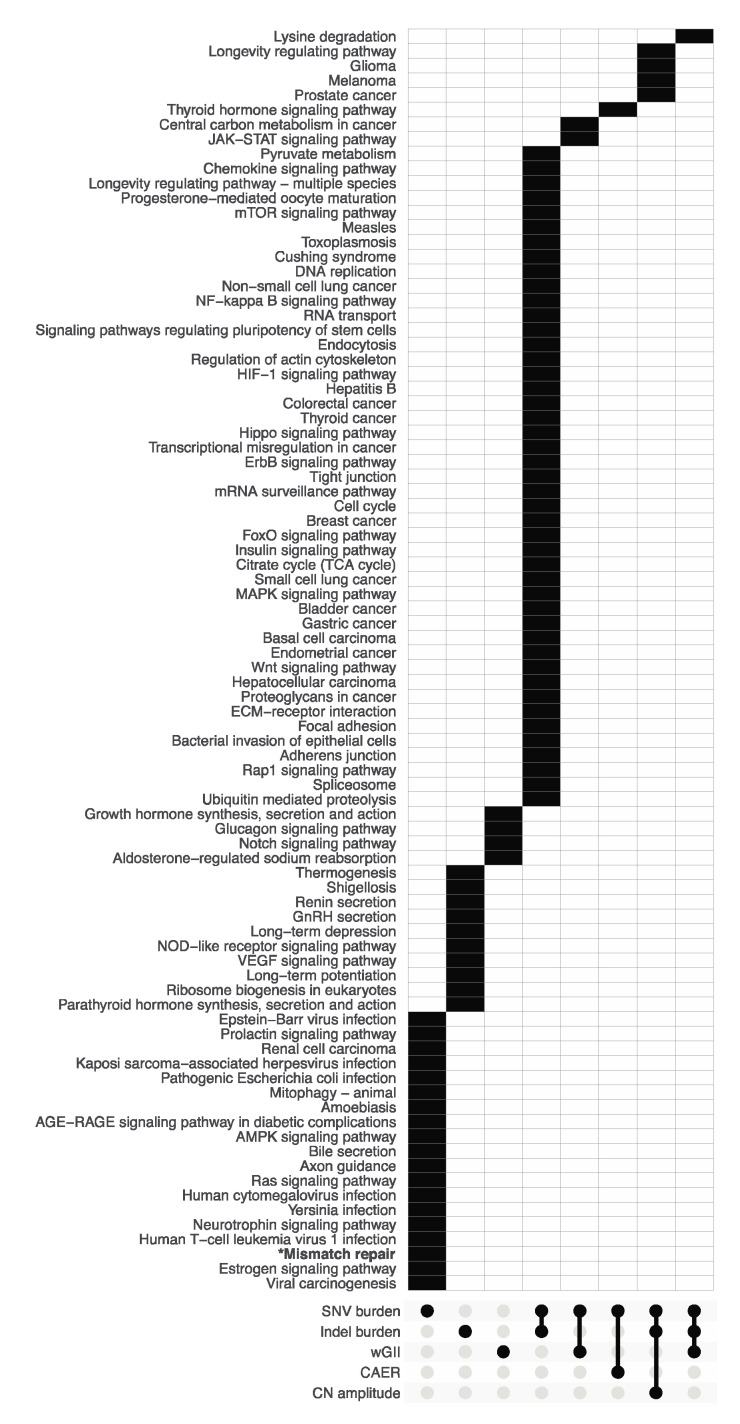
“Mismatch repair” is significantly associated with SNV burden. The UpSet plot displaying the sets of results of pathway enrichment analyses on genes associated with each of the 5 quantitative metrics.

**Figure 4 biomedicines-08-00574-f004:**
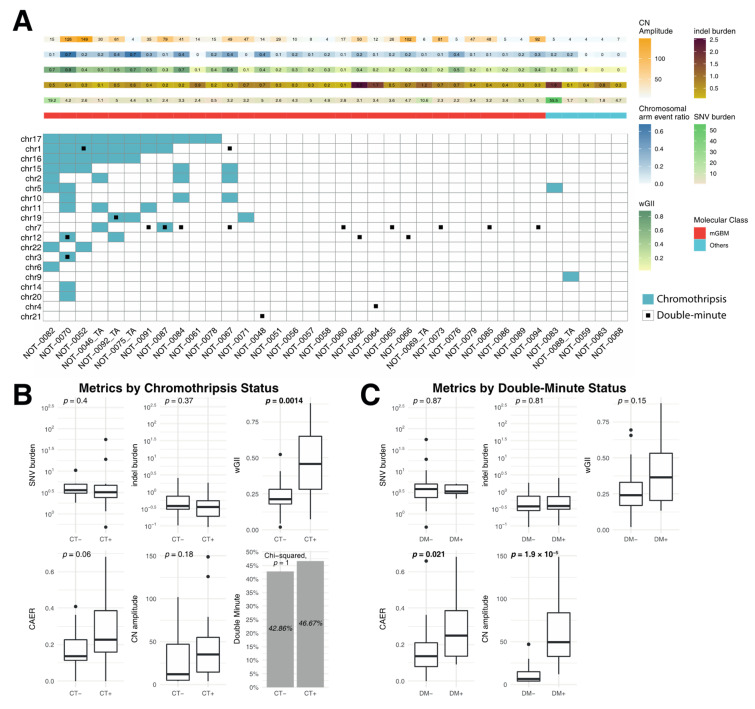
Chromothripsis and double minute events are associated with copy-number alteration. (**A**) A waterfall plot displaying the overview of chromothripsis and double minute events in primary tumors. (**B**) Comparison of metrics according to chromothripsis status. Cases with chromothripsis had significantly higher frequency of copy-number alterations (wGII). (**C**) Comparison of metrics according to double minute status. Cases with double minutes had significantly higher copy-number amplitudes and chromosomal arm event ratios. Bold *p* values indicate statistical significance (*p* < 0.05).

**Figure 5 biomedicines-08-00574-f005:**
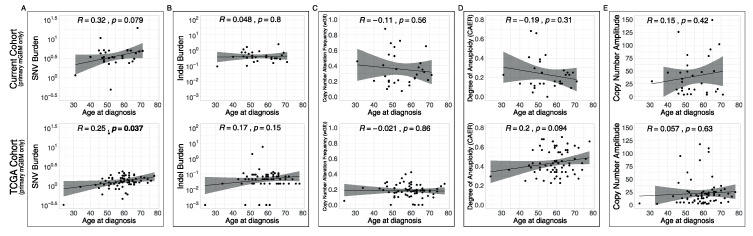
Only SNV burden correlates with age. (**A**–**E**) Scatter plots between age at diagnosis and SNV burden (**A**), indel burden (**B**), wGII (**C**), CAER (**D**) and CN amplitude (**E**) in the current cohort (upper) and the TCGA cohort (lower). Bold *p* value indicates statistical significance (*p* < 0.05).

**Figure 6 biomedicines-08-00574-f006:**
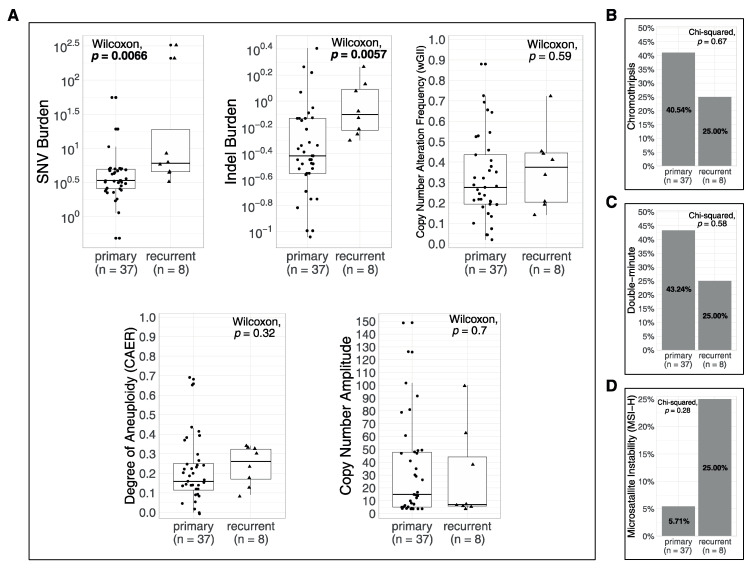
Only SNV burden and indel burden increase with recurrences. (**A**) Comparison of quantitative metrics between all primary and all recurrent cases. (**B**) Comparison of chromothripsis prevalence between primary and recurrent cases. (**C**) Comparison of double minute prevalence between primary and recurrent cases. (**D**) Comparison of MSI prevalence between primary and recurrent cases. Bold *p* values indicate statistical significance (*p* < 0.05).

**Figure 7 biomedicines-08-00574-f007:**
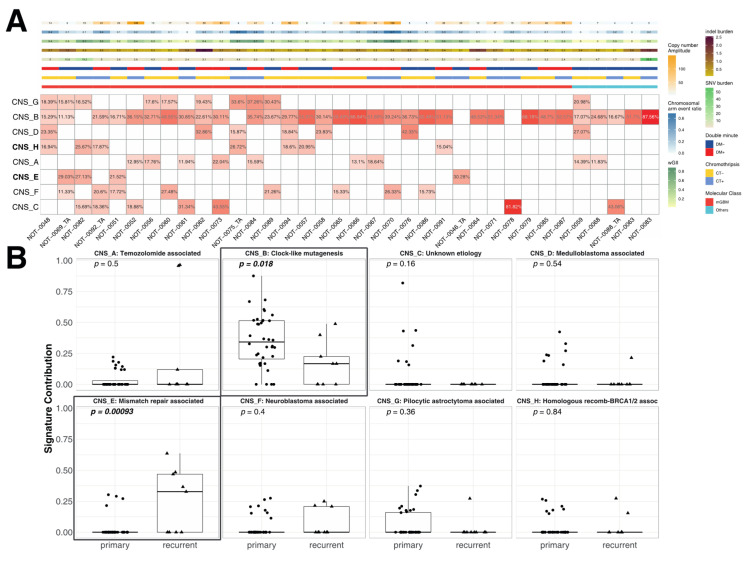
Contributions of clock-like and mismatch repair deficiency-associated signatures differ between primary and recurrent tumors. (**A**) A heatmap of central nervous system (CNS)-associated mutational signatures in primary tumors (mismatch-repair-associated signature CNS-E and homologous-recombination-associated signature CNS-H are marked in bold). Clock-like signature CNS-B was the most common signature. (**B**) Comparison of signature contributions between primary and recurrent tumors. Mismatch-repair-associated signature CNS-E was significantly higher in recurrent tumors. Bold *p* values indicate statistical significance (*p* < 0.05).

**Table 1 biomedicines-08-00574-t001:** Characteristics of the patients and tumors analyzed in the study. “Primary status” indicates whether the tumor is a “primary” tumor or a “recurrent” tumor. “Gender” indicates the gender of the patient: “M” for male and “F” for female”. “Sample type” indicates whether the tumor sample is a fresh frozen tissue sample (LiN2) or Formalin-Fixed Paraffin-Embedded (FFPE) sample. “*ATRX*” indicates the presence of a somatic mutation in the gene *ATRX*, “WT” indicates wild-type, whereas “MUT” indicates mutated *ATRX*. “*TERT*” indicates the *TERT* promoter mutation status, “WT” indicates wild-type, “C228” and “C250” indicate somatic mutations at the given genomic positions. “H3” indicates any somatic mutations in the gene *H3F3A*, “WT” indicates wild-type, otherwise the protein alteration is presented. “*EGFR* amplification” indicates whether the gene *EGFR* is amplified (“amplification”) or not (“copy neutral”). “7+/10−” indicates whether both whole chromosome 7 amplification and whole chromosome 10 deletion is observed (TRUE) or not (FALSE). “Molecular Subset” indicates the molecular subset of the tumor: either molecularly-defined Glioblastoma (“mGBM”) or “Others”.

Patient ID	Analysis ID	Primary Status	Gender	Age at Initial Presentation	Predominant Localization	Sample Type	Pathological Diagnosis	Grade	*ATRX*	*TERT*	*H3*	*EGFR* Amplification	7+/10−	Other Putative Drivers	Molecular Subset
NOT-0046	NOT-0046_TA	primary	M	31	thalamic	FFPE	Glioblastoma, *IDH* wild-type	IV	MUT	WT	WT	amplification	FALSE		mGBM
	NOT-0046_TB	recurrent	M	31	thalamic	LiN2	Glioblastoma, *IDH* wild-type	IV	MUT	WT	WT	amplification	TRUE		mGBM
NOT-0047	NOT-0047	recurrent	M	49	parietal	LiN2	Glioblastoma, *IDH* wild-type	IV	WT	WT	WT	amplification	TRUE		mGBM
NOT-0048	NOT-0048	primary	F	45	frontal	LiN2	Glioblastoma, *IDH* wild-type	IV	WT	WT	WT	amplification	TRUE		mGBM
NOT-0051	NOT-0051	primary	M	48	temporal	FFPE	Glioblastoma, *IDH* wild-type	IV	WT	C228	WT	amplification	TRUE		mGBM
NOT-0052	NOT-0052	primary	F	65	hippocampus	FFPE	Anaplastic astrocytoma, *IDH* wild-type	III	WT	C228	WT	amplification	TRUE		mGBM
NOT-0054	NOT-0054	recurrent	F	40	frontal	LiN2	Glioblastoma, *IDH* wild-type	IV	WT	C250	WT	amplification	TRUE		mGBM
NOT-0056	NOT-0056	primary	F	67	hippocampus	LiN2	Glioblastoma, *IDH* wild-type	IV	WT	C228	WT	amplification	TRUE		mGBM
NOT-0057	NOT-0057	primary	F	62	temporal	LiN2	Glioblastoma, *IDH* wild-type	IV	WT	C228	WT	amplification	TRUE		mGBM
NOT-0058	NOT-0058	primary	F	71	frontal	FFPE	Glioblastoma, *IDH* wild-type	IV	WT	C228	WT	amplification	TRUE		mGBM
NOT-0060	NOT-0060	primary	M	48	hippocampus	FFPE	Glioblastoma, *IDH* wild-type	IV	WT	C250	WT	amplification	TRUE		mGBM
NOT-0061	NOT-0061	primary	M	55	occipital	FFPE	Glioblastoma, *IDH* wild-type	IV	WT	C250	WT	copy neutral	FALSE		mGBM
NOT-0062	NOT-0062	primary	M	66	frontal	FFPE	Glioblastoma, *IDH* wild-type	IV	WT	C250	WT	amplification	TRUE		mGBM
NOT-0064	NOT-0064	primary	F	48	frontal	LiN2	Glioblastoma, *IDH* wild-type	IV	WT	C228	WT	amplification	TRUE		mGBM
NOT-0065	NOT-0065	primary	M	59	occipital	LiN2	Glioblastoma, *IDH* wild-type	IV	WT	C228	WT	amplification	FALSE		mGBM
NOT-0066	NOT-0066	primary	F	69	frontal	LiN2	Glioblastoma, *IDH* wild-type	IV	WT	C228	WT	amplification	TRUE		mGBM
NOT-0067	NOT-0067	primary	F	51	parietal	LiN2	Glioblastoma, *IDH* wild-type	IV	WT	WT	WT	amplification	TRUE		mGBM
NOT-0069	NOT-0069_TA	primary	M	46	parietal	FFPE	Glioblastoma, *IDH* wild-type	IV	WT	C228	WT	amplification	TRUE		mGBM
	NOT-0069_TB	recurrent	M	46	parietal	LiN2	Glioblastoma, *IDH* wild-type	IV	WT	C228	WT	amplification	TRUE		mGBM
NOT-0070	NOT-0070	primary	M	46	temporal	FFPE	Glioblastoma, *IDH* wild-type	IV	WT	C250	WT	copy neutral	TRUE		mGBM
NOT-0071	NOT-0071	primary	M	47	multifocal	LiN2	Glioblastoma, *IDH* wild-type	IV	WT	WT	WT	amplification	FALSE		mGBM
NOT-0073	NOT-0073	primary	M	51	frontal	LiN2	Glioblastoma, *IDH* wild-type	IV	WT	C228	WT	amplification	TRUE		mGBM
NOT-0076	NOT-0076	primary	M	51	thalamus	LiN2	Glioblastoma, *IDH* wild-type	IV	WT	C228	WT	amplification	TRUE		mGBM
NOT-0078	NOT-0078	primary	F	52	frontal	LiN2	Diffuse astrocytoma, WHO grade II, *IDH* wild-type	II	MUT	WT	WT	amplification	FALSE		mGBM
NOT-0079	NOT-0079	primary	M	40	parietal	LiN2	Glioblastoma, *IDH* wild-type	IV	WT	C228	WT	amplification	TRUE		mGBM
NOT-0082	NOT-0082	primary	M	68	gliomatosis	FFPE	Anaplastic astrocytoma, WHO grade III, *IDH* wild-type	III	WT	C228	WT	amplification	FALSE		mGBM
NOT-0084	NOT-0084	primary	M	54	parietal	LiN2	Glioblastoma, *IDH* wild-type	IV	WT	WT	WT	amplification	FALSE		mGBM
NOT-0085	NOT-0085	primary	M	59	frontal	LiN2	Glioblastoma, *IDH* wild-type	IV	WT	C250	WT	amplification	TRUE		mGBM
NOT-0086	NOT-0086	primary	M	53	temporal	FFPE	Glioblastoma, *IDH* wild-type	IV	WT	C250	WT	amplification	TRUE		mGBM
NOT-0087	NOT-0087	primary	M	63	frontal	FFPE	Glioblastoma, *IDH* wild-type	IV	WT	C250	WT	amplification	TRUE		mGBM
NOT-0089	NOT-0089	primary	M	62	parietal	LiN2	Glioblastoma, *IDH* wild-type	IV	WT	C228	WT	copy neutral	FALSE		mGBM
NOT-0091	NOT-0091	primary	F	48	frontal	FFPE	Glioblastoma, *IDH* wild-type	IV	WT	C228	WT	amplification	TRUE		mGBM
NOT-0092	NOT-0092_TA	primary	M	51	frontal	LiN2	Glioblastoma, *IDH* wild-type	IV	WT	WT	WT	amplification	FALSE		mGBM
	NOT-0092_TB	recurrent	M	51	frontal	FFPE	Glioblastoma, *IDH* wild-type	IV	WT	WT	WT	amplification	FALSE		mGBM
NOT-0094	NOT-0094	primary	M	64	frontal	FFPE	Glioblastoma, *IDH* wild-type	IV	WT	C228	WT	amplification	TRUE		mGBM
NOT-0075	NOT-0075_TA	primary	F	49	cerebellar	LiN2	Anaplastic astrocytoma, *IDH* wild-type	III	MUT	WT	WT	amplification	TRUE		mGBM
	NOT-0075_TB	recurrent	F	49	cerebellar	LiN2	Glioblastoma, *IDH* wild-type	IV	MUT	WT	WT	copy neutral	FALSE		mGBM
NOT-0059	NOT-0059	primary	F	76	frontal	FFPE	Glioblastoma, *IDH* wild-type	IV	WT	WT	WT	copy neutral	FALSE	*SETD2* (Y2523)	Others
NOT-0063	NOT-0063	primary	M	37	gliomatosis	LiN2	Glioblastoma, *IDH* wild-type	IV	MUT	WT	G34R	copy neutral	FALSE		Others
NOT-0068	NOT-0068	primary	M	64	gliomatosis	LiN2	Diffuse astrocytoma, *IDH* wild-type	II	MUT	WT	WT	copy neutral	FALSE	*SETD2* (A2553T)	Others
NOT-0083	NOT-0083	primary	F	41	corpus callosum	FFPE	Glioblastoma, *IDH* wild-type	IV	MUT	WT	WT	copy neutral	FALSE	*SETD2* (R2040*; R2510H)	Others
NOT-0088	NOT-0088_TA	primary	F	34	cerebellar	LiN2	Anaplastic astrocytoma with piloid features, *IDH* wild-type*	III	WT	WT	WT	copy neutral	TRUE	*FGFR1* (K567E; V742M)	Others
	NOT-0088_TB	recurrent	F	34	cerebellar	LiN2	Glioblastoma, *IDH* wild-type	IV	WT	WT	WT	copy neutral	FALSE	*FGFR1* (K567E; V742M)	Others
NOT-0090	NOT-0090_TA	primary	M	28	frontal	LiN2	Glioblastoma, *IDH* wild-type	IV	MUT	WT	WT	copy neutral	FALSE	*SETD2* (KQ583fs)	Others
	NOT-0090_TB	recurrent	M	28	frontal	LiN2	Glioblastoma, *IDH* wild-type	IV	MUT	WT	WT	copy neutral	FALSE		Others

**Table 2 biomedicines-08-00574-t002:** Characteristics of the metrics analyzed in this study. “CNS” indicates central nervous system. “CNS_“ “A”, “B”, “C”, “D”, “E”, “F”, “G” and “H” indicate different CNS-related mutational signatures.

Metric	Assessed Genomic Alteration	Possible Mechanisms
**SNV Burden**	Frequency of single nucleotide variations (SNV)	DNA damage repair deficiency, Polymerase errors, *APOBEC* mutagenesis in hypermutated/ultra-mutated tumors
**Indel Burden**	Frequency of short insertion deletions (indels)	Polymerase slippage, Non-homologous End Joining (NHEJ), hairpin loops
**Weighted Genome Instability Index (wGII)**	Frequency of copy number variation (CNV) events	Double stand breaks, NHEJ, Mitotic nondisjunction, Chromosomal instability, Break-Fusion Bridge (BFB) cycles, Chromothripsis
**Chromosomal Arm Event Ratio (CAER)**	Number of chromosomal arm amplifications or deletions (excludes X, Y): Aneuploidy	Mitotic nondisjunction, Chromothripsis
**Copy number amplitude**	Maximum number of amplifications	Extrachromosomal minutes and chromothripsis in cases with high level amplification (>10)
**Chromothripsis**	Massive, clustered, single chromosomal rearrangements	Mitotic nondisjunction
**Double Minutes**	circularization of double-stranded DNA, resulting in highly amplified genes	Chromothripsis, gene amplification, NHEJ
**Microsatellite instability (MSI)**	Length variation in microsatellite repeats	Mismatch Repair (MMR) deficiency
**Mutational Signatures**	Mechanisms underlying single nucleotide variations	CNS_A: Temozolomide-associated CNS_B: Clock-like mutagenesis CNS_C: Unknown etiology CNS_D: Medulloblastoma-associated CNS_E: Mismatch-repair-associated CNS_F: Neuroblastoma-associated CNS_G: Pilocytic astrocytoma-associated CNS_H: Homologous recombination-*BRCA1/2*-associated

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
