# Peer review of "Mutations and Copy Number Alterations in *IDH* Wild-Type Glioblastomas Are Shaped by Different Oncogenic Mechanisms"

_biomedicines, 2020, doi:10.3390/biomedicines8120574_

Round 1

Reviewer 1 Report

This work is aimed at determining whether there are associations between the burden of various genomic alterations and mechanisms that could be causing cancer genome instability and chromosome alterations. The authors studied gliomas lacking IDH mutations (37 primary tumors and 8 recurrences) by whole-exome sequencing. Tumors were sub-grouped into molecularly defined gliomas -mGBM- (those having TERT promoter mutations and/or chromosome 7 amplification/chromosome 10 deletion and/or EGFR-amplification) and ‘other’ gliomas lacking those molecular events. Using the sequencing data, the authors quantified the burden of five different genome alterations: SNV, Indels, and CNA as well as the maximal amplitude of CNA and the ratio of chromosomal arm events.

The conclusions include that SNV burden are: (1) positively correlate with increase age of diagnosis, (2) is higher in mGBM than other gliomas and also (2) in recurrent tumors as compared with primary tumors, and (3) may be caused by mismatch repair deficiencies. I find several issues that should be addressed:

  1. The data on the authors’ cohort actually shows no differences between mGBM and ‘other’ gliomas. This is not validated in the TCGA data which do yield differences. This inconsistency does not allow to reach such a categorical conclusion as it is stated in the manuscript, even in the abstract section.
  2. Correlation with age is indicated as variation through time which is a confusing term.
  3. Pathway enrichment analysis of metric-associated somatic variants was performed to unravel the possible mechanisms underlying each metric. In figure 3 legend, authors state that “Mismatch repair” is significantly associated with SNV burden. This is not explained in the results section. In addition, “Mismatch repair” pathway is not among the enriched pathways showed in Figure 3. Please clarify and show the data that support this conclusion.

Another conclusion is that mGBM and other gliomas differ in CNA. Given that mGBM are selected as tumors with 7/10 genotype (the most frequent CNA as shown in Figure S1), the association is a rather expectable result. Please clarify this.

A correlation between CT and CNA burden is interpreted as a cause-effect relationship. This should be discussed and proven. The same for DM and CAER or CN amplitude.

Other concerns are:

  • The first paragraph of the result section should indicate whether that analysis included all tumor types (mGBM and others?; primary + recurrences?). Only the authors’ cohort or also the TCGA data?
  • The authors examined the genes associated with each metric and show the data as supplementary material but do not highlight whether any significant discovery is achieved using this approach. They only highlight the association between mismatch repair pathway and SNV which, as mentioned above, it is a result not showed in Figure 3.
  • Please clarify also the data on clustering analysis: what means that SNV burden and indel burden, associated with mutational processes (what do you mean by “mutational processes”?) and (2) CN amplitude, DM, CT, wGII and CAER, associated with SCNA-related mechanisms (Could you define SCNA-related mechanisms?).
  • Why only age of diagnosis is used for clinical-genetic association analysis? What about other clinical data?
  • Sample size (specially the so-called “other” gliomas is very reduced. Please consider increasing sample size.

Author Response

Thank you for your valuable review

Reviewer 2 Report

Interesting aspects regarding genetics of IDH wild type Glioblastomas. I think what could be improved though in the Discussion and conclusion would be to highlight the potential clinical impact your findings could have.

Author Response

Thank you for your valuable review. We expanded the Conclusions section to reflect the potential clinical impact of the study:

"We hope that this work contributes to the deeper understanding of the tumor biology underlying IDH-WT tumor entities will allow for better characterization of these tumors to better understand clinical behavior and eventually for developing novel treatment strategies."

Reviewer 3 Report

In the work, „Mutations and Copy Number Alterations in IDH3 Wild Type Glioblastomas are Shaped by Different Oncogenic Mechanisms“, the autors Ülgen et al. present a detailed and well structered work analysing the genetic landscape of IDH wild type glioblastomas, however, opening interesting perspectives with regard ro cancer in general.

The work is full of new ideas and insight, however some aspects should be improved.

The authors report several subtypes ogf glioblastoma, however, do not report primary versus secondary GBM, which is associated to the IDH status, however, this should be mentioned.

Further, the differentiation between mGBM and „others“ they perform should be explained more in detail in order to appear not just arbitrary.

In fact, although the work contains an enormous amount of data and their diligant anlysis, some entities still are singular, not allowing any general conclusions, as anaplastic astrocytoma. So these limitations should be described more in detail.

IN conclusion, the work is full of interesting insight, however, some details should still be improved.

Author Response

thank you for your valuable review and comments

Reviewer 4 Report

Ülgen et al present a paper entitled “Mutations and Copy Number Alterations in IDH Wild Type Glioblastomas are Shaped by Different Oncogenic Mechanisms”

Introduction should be expanded; in particular the author should at least introduce IDH gene considering the at pag 1, line 63, they started to talk about IDH-WT pts. This is important for non-expert readers.

Add a statement “ the aim of this study was”

Materials and Methods

“Thirty-nine adult patients…..were included. IDH-mutant gliomas (astrocytomas and oligodendrogliomas) as well as diffuse midline gliomas H3-K27M mutant were excluded.” What is the final number of pts?

Also: “45 tumors were studied”. Please modify this sentence whit “45 tumor samples from 39(??) were studied”. With this sentence the authors respond also to the previous comment.

Please double check the number of patients even in the result section. It look like that At pag 8, line 167 there are different numbers

Table 1 is very difficult to read. Please increase the font size

Figure 1: please increase the font size of the p val

Also figure 7a is too small and it is impossible to read. Can the authors use the horizontal page option?

In paragraph 4.6 add the small size cohort among the study limits

In general the paper is interesting and I think it is suitable to be published after revision. In some points the manuscript is difficult to follow and in my opinion the authors could simplify it. The text of alll the figures is very small and almost impossible to read

Ülgen et al present a paper entitled “Mutations and Copy Number Alterations in IDH Wild Type Glioblastomas are Shaped by Different Oncogenic Mechanisms”

Introduction should be expanded; in particular the author should at least introduce IDH gene considering the at pag 1, line 63, they started to talk about IDH-WT pts. This is important for non-expert readers.

Add a statement “ the aim of this study was”

Materials and Methods

“Thirty-nine adult patients…..were included. IDH-mutant gliomas (astrocytomas and oligodendrogliomas) as well as diffuse midline gliomas H3-K27M mutant were excluded.” What is the final number of pts?

Also: “45 tumors were studied”. Please modify this sentence whit “45 tumor samples from 39(??) were studied”. With this sentence the authors respond also to the previous comment.

Please double check the number of patients even in the result section. It look like that At pag 8, line 167 there are different numbers

Table 1 is very difficult to read. Please increase the font size

Figure 1: please increase the font size of the p val

Also figure 7a is too small and it is impossible to read. Can the authors use the horizontal page option?

In paragraph 4.6 add the small size cohort among the study limits

In general the paper is interesting and I think it is suitable to be published after revision. In some points the manuscript is difficult to follow and in my opinion the authors could simplify it. The text of alll the figures is very small and almost impossible to read

Author Response

Thank you for your valuable review and comments

Round 2

Reviewer 1 Report

The authors have provided, to a large enough extent, responses to all my comments and suggestions. 

Reviewer 4 Report

the authors revised the paper according my comments.

- please correct the sentence:

“45 tumors from 39 patients were studied" in “45 tumor specimens (or 45 tumor samples) from 39 patients were studied"..

- the same in the results

“For analyses, 45 IDH-WT diffuse
glioma tumor specimens from 39 patients were used…”

technically you did not use the entire tumor but only a part of that (i suppose )